# 'I Respect You but I Am Not Willing to Be You': Critical Reflections of Western Teaching of Social Work to Students in China—What Can be Learned Both Ways?

**Hilary Gallagher** [1,2,3]**, Liuqing Yang** [3,4] **and Jianqiang Liang** [1,2,3,]*

[1] School of Human Services and Social Work, Griffith University, Gold Coast, QLD 4215, Australia; h.gallagher@griffith.edu.au
[2] Menzies Health Institute Queensland, Griffith University, Gold Coast, QLD 4215, Australia
[3] Australia China Social Work Research Centre, Central China Normal University, Wuhan 430000, Hubei, China
[4] School of Sociology, Central China Normal University, Wuhan 430000, Hubei, China; pyuanhl00178@163.com
[*] Correspondence: j.liang4@griffith.edu.au; Tel.: +61-7-3382-1161

**Abstract:** Staff from a Western University annually travel to China to teach social work students at a Chinese University, providing a rich opportunity to share ideas and knowledge about values and practices in social work. One common point of tension that arises each year is how to teach critical reflection whilst considering differences between Eastern and Western ways of knowing and doing. This article is based on email conversations between one Australian lecturer and one Chinese student, containing their discussions on not just critical reflection but also of various key social work topics in China such as social worker's salary, social work as a profession and using empathy. The student questioned social work in an authentic and practical manner; while the lecturer responded with examples and reflections as a role model of critical reflective thinking and practice in the Chinese context. While such letters of exchange only reflect the particular points of view of the lecturer and the student, much can still be learned about current issues and debates in both countries. The insights given raise many questions about the implications and benefits for sensitively teaching social work across East/West contexts whilst trying to develop anti-colonial social work educational approaches.

**Keywords:** critical reflection; higher education; social work; China; email

## 1. Introduction

Social work initially started on the Chinese Mainland in the 1920s. However, professional social work and tertiary social work education were suspended by the Chinese Government from early 1950s to late 1980s. Social work was reintroduced to university programs in the late 1980s and early 1990's due to economic reform. Currently, social work education in China is experiencing growth in order to meet the needs of the "The Medium-to-Long Term Talents Development Plan 2010–2020," with the number of professionally qualified social workers expected to rise to nearly one and a half million by 2020 (Guo 2017; Mo et al. 2019).

Social work has originated from a Western social, religious and economic background, with some scholars arguing that it is impossible to apply Western social work principles in China (Cheung and Liu 2004). The development of social work in China needs to be rooted in the social and economic systems present in China, such as the family and be reflected in social work education programs (Cheung and Liu 2004). To professionalize the discipline, social work values such as individual rights, dignity and

equality originating from Western capitalist countries, need to be adapted to the political context in China (Yip 2007). For example, the term empowerment, when applied in China, means developing the potential for a client to be responsible, caring and self-respecting, for the betterment of their family and community. This also includes the role of families in taking care of individuals, including people with disabilities (Yip 2007). This definition differs from Western contexts and demonstrates how terms have been adapted to fit in with the values inherent in Chinese culture.

Cross-cultural teaching of social work subjects in China has become an emerging issue. This paper is not based on a traditional scientific study, rather a critical reflection on Western teaching from a Western social work teacher and her email conversations with a local, Chinese-born social work undergraduate student. Liuqing (Molly) and Hilary met each other when Hilary taught a critical reflective practice social work course in China. Currently Chinese social work degrees are four years but critical reflection and field education content is less formal than in Australia. There are placement opportunities in community centres or non-government organizations but these tend to be short and without guided supervision and learning. Attending classes provided by Australian lecturers can provide some transformative learning opportunities for students, such as Molly. Equally the discussion and critical reflection of students, like Molly, provides a rich learning environment for Australian lecturers, such as Hilary, to reflect on the global profession of social work and the practicalities and ethics of teaching a profession with colonial history, in other countries.

## 2. Method

This article is based on the personal knowledge, thoughts, questions and reflections of Molly and Hilary over a three-week learning and teaching period in China. They tried to seek clarification within the literature to validate their thoughts and Hilary consulted with Jianqiang (Joe), a Chinese-born social work lecturer, as a critical peer, to discuss issues of teaching and learning in the Chinese context. Bassot (2016) describes reflective practice as having many facets, including the space to analyse and review practice, construct professional knowledge and develop better understanding. However, it must be acknowledged that critical reflection is a process that questions our underlying beliefs and assumptions (Fook and Askeland 2007). It is therefore subjective and as such this article can only offer one lens through which to understand the issues discussed. Experiential learning is used in social work field education and many taught courses (Wayne et al. 2010). Hilary and Molly learned much from each other's conversations but in terms of reaching a new understanding, this continues to be a work in progress, in a changing cultural landscape.

## 3. Email Conversations

The emails from the student Molly are in normal font, while the emails from the lecturer, Hilary, are in italics.

Molly wrote, "Dear Hilary, I'm deeply aware that social work is a major that affects people's lives. I think the students who study this major should understand this but many students who major in social work do not regard social work as their ideal career. I have three questions I would like to discuss with you: (1) the wage of social workers, (2) professional issues, (3) the problem of empathy."

Hilary wrote, "*Dear Molly, Thank you. These are very important issues with no simple answers. There are no right or wrong answers for social work in China at this point in time, as it is an emerging and developing field. I will try to answer your thoughts with my own reflections, as best as I can.*

*Academics from our two universities are aware that these are current issues which is one of the reasons we see our partnership as very important. Social work is about changing people's lives and society for the better, whatever that looks like in each individual country (Chenoweth and McAuliffe 2017). It is not and cannot be the same in China and Australia as we have different ways of being and a different history and culture. For example, you and I might show empathy in different ways (Lin and Appleton 2018) or see a student and supervisory*

*relationship as different (Mo et al. 2019). However, the underlying principles of justice, professionalism and humanity are similar.*

*Australia and China are both members of the International Federation of Social Work (IFSW International Federation of Social Workers) which promotes human rights and social justice principles. The website describes social work as a "game changer"! Change is not a short-term goal and may be for your generation, social work is not a viable profession yet. Guo (2017) describes career barriers you might experience as being limited professional roles, limited recognition of social work as being a profession, low wages and a lack of a structured career pathway. However, if we do not work now, to develop social work as a profession, then it may never happen.*

*Guo (2017) research did highlight the need for government and social institutions to think about increasing wages, social recognition and job opportunities; but it sounds as if this could take time. On a positive note, there were nearly 80 posters, papers and presentations from China at the Social Work Education and Social Development Conference (Garrett 2019) that I went to in Dublin in 2018. In the meantime, social work academics still see studying social work as important because critical thinking and a knowledge of people and communities, is important in all areas of work (especially the humanities) so by teaching social work now, graduates may make changes in their professions to encourage the promotion of social work principles, valuing people and a harmonious society. For example, teaching critical reflection offers students the opportunity to learn skills to develop self-awareness, reflective thinking, critical analysis and the ability to apply theory to practice (Reimer and Whitaker 2019). We need these skills to be able to consider and understand the local social, economic, cultural and political context."*

### 3.1. The Wage of Social Workers

Molly wrote, "Dear Hilary, here is my first issue: in the Chinese Mainland, social workers' wages are generally low. In today's society, wage is an important criterion for judging a person's social status. High knowledge input does not bring high-income returns, which makes many students question their careers, that is, whether it is worth working in social work.

This picture (Figure 1) shows the average wage of social workers in Wuhan (Zhiyouji 2019). Assuming that 1000 yuan is used for renting accommodation and 2000 yuan for daily life, I could potentially save 9000 yuan a year. However, my major tuition and accommodation fees for one year of university is 26,120 yuan. This means that I would need 104,480 yuan after four years of study (not counting my minor tuition and living expenses). If I work as a social worker in Wuhan, I would not be able to pay back my tuition fees, even after ten years of working. Housing prices are also expensive in Wuhan. I might be able to buy two square meters, if I did not eat or drink for a year! Even if I worked until retirement, I could not afford to buy a house.

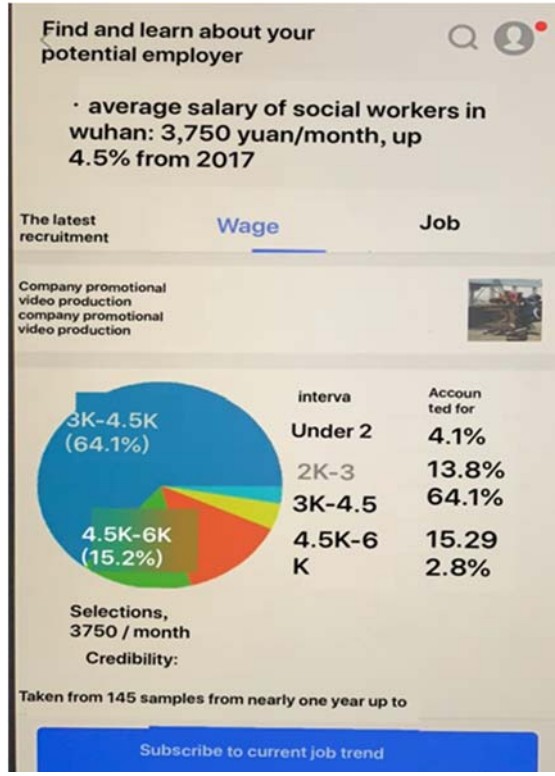

**Figure 1.** Average Salary of Social Workers in Wuhan, May 2019.

It's hard to imagine what kind of educational resources I could provide for my children under such living conditions and I would not be able to pay for them to study abroad. I am very grateful to my parents for paying my high tuition fees to give me a chance to study at university. I am also very grateful to the university for providing me with an opportunity to exchange and study abroad and the opportunity to sit here and listen to your lectures, which is based on a lot of money."

Hilary wrote, "*Dear Molly, I remember reading about how organisational culture influences how we regard and value, wealth and knowledge (De Long and Fahey 2000). I think that extends to our wider cultural viewpoint. I believe Australia is lucky, as enough (maybe not high amounts) of wealth can give a good life, so social work is viable for people who do not value wealth as highly as knowledge/changing society for the better. Social work is not the most highly paid profession in Australia, but people can pay off tuition fees, eat and afford a home, especially if there are two wage earners in the family.*

*Wages and the cost of living is a huge problem for social work in China. Shenzhen was one of the pilot areas for developing social work jobs and in fact, the Shenzhen Social Work Association said that poor wages are one of the two top reasons as to why social workers leave their jobs (Jiang et al. 2019). Students whose families can afford education, can study social work but as graduates cannot afford to work as one. I think low wages are connected to your question about professionalism as well. If the job is not valued by society, then it is difficult to increase remuneration (Jiang et al. 2019). My thoughts are that while social work is an emerging field in China, perhaps management and government jobs are an important career route, then your social work studies can help in discussing/developing new policies and legislation.*

*Your government has many plans to develop social governance, social welfare systems and maintain social stability and recognises the importance of social work (Guo 2017; Li et al. 2012). Once social work is more recognised as a profession then social workers will hopefully gain more pay. I think you are at the forefront of many changes but I'm not sure how long they will take. Academics have ethical debates about the ethics of teaching social work when there is not yet a developed profession to employ you. Particularly, when the current Chinese social work professional qualification is not dependent on a tertiary education (Li et al. 2019),*

*which clearly has implications for professionalisation and wages. However, quality education and developing a professional identity is so important, when looking to change in the long-term (Lei et al. 2019)."*

Molly wrote, "Dear Hilary, sometimes I wonder if low wages and poor living conditions can foster social workers' empathy? Occupations like teachers, police officers, doctors and social workers can't appear to care about their salaries, because society has put a noble label on these occupations. You serve society and you are a good person, so you shouldn't care about your own gains and losses. This kind of social desirability makes some people suppress their inner desire for high salary (there are actually some people who really think their salary is enough but not everyone). It also seems that some of the noble professions have more status and remuneration opportunities than other careers like social work (Blumenthal and Hsiao 2015)."

Hilary wrote, *"Dear Molly, I also wonder how sustainable empathy is, in terms of wages. If social workers are disadvantaged themselves, then how can they support others in the long-term without experiencing compassion fatigue (Radey and Figley 2007)? In some ways, life experiences may enhance empathy but may also limit capacity and resources to make changes. Social work is re-emerging as a profession in China but there are already studies about burnout and turn-over related to a lack of organisational supports (Tang et al. 2017). This is another reason why social work needs to be a respected profession (Lei et al. 2019).*

*It is easy to think your salary is enough when you can pay for food, a home and educate your children, which is a reason why social work is fortunate in Australia. However, social workers still do not earn as much as doctors and lawyers. Social work is a value-based profession (Chenoweth and McAuliffe 2017) and I think many social workers chose to live a personal life based on their values, such as sustainability and equity. This can often take away the need to be rich. For example, I volunteer each week with children. I take them on activities like camping, rock climbing bush walking and abseiling. My own children are also a part of the group. We have fun that is not based on money but rather in the environment and outdoors."*

## 3.2. Professional Issues

Molly wrote, "Dear Hilary, I also think about the second issue below. I think the major of social work is simpler than sociology, psychology, mathematics, physics, chemistry, biology and philosophy. As an interdisciplinary subject, social work is involved in many fields but not in-depth.

For example, Alzheimer's disease, which is studied by psychologists and doctors. At present, the pathogenesis of Alzheimer's disease is not clear (Tao et al. 2019). I think it may also be related to early depression, schizophrenia, paranoid psychosis and factors like an improper diet (such as alcohol), water containing aluminium and the accumulation of neurotoxins in the body. Even if social workers have some knowledge about a disease, they cannot know all the up to date research and cannot give direct treatment, as they need to refer clients to professional medical personnel for systematic examinations and medication."

Hilary wrote, *"Dear Molly, if you study one of the major's you mentioned, you may still only understand a small part of your major in a lot of depth. For example, in my first degree, I majored in optoelectronics and laser physics. However, this is only one, tiny part of the field of physics, ideally aluminium. Social work graduates take time to build social work expertise on graduation in order to transition to become a competent practitioner (Hunt et al. 2017). As a social worker, it may seem like you have a shallow understanding of a wide area but as your career progresses, you become much more knowledgeable in a specific area, for example, mental health, disability, policy or domestic violence. Social workers are also trained to see the wide range of impacts of these factors on individuals, groups and communities. Systems theory helps to explain the interconnectedness between, for example, a person, their circumstances and external systems. Social work knowledge can then help social workers advise specialists on how different systems will affect each other and work more holistically to facilitate change (Chenoweth and McAuliffe 2017).*

*Social workers may not know about medication or treatment but they can discuss which treatment may support the person the best. Currently with regards to Alzheimer's, some research is suggesting the cause as*

*proteins in the brain (Tao et al. 2019) and practice wisdom suggests that stress may exacerbate the symptoms. Social workers can discuss with gerontologists the effect of, for example, whether a person should undertake a clinical trial or whether the stress of ageing-related operations may exacerbate the symptoms of Alzheimer's and help the client and specialist decide whether operations are necessary."*

Molly wrote. "Dear Hilary, the biggest role of front-line social workers is to link resources, which has led some outsiders to question the professionalism of social work (Zhang et al. 2019). That is, what is the difference between volunteer work and social work?"

Hilary wrote, "*Dear Molly, this is a very important and contemporary question. As you said, Zhang et al. (2019) discuss the importance of developing professional social work and competency-based practice using educational tools such as collaborative, experiential and problem-based learning and critical reflective practice. Developing competency-based field education is another way to highlight the complexities of social work practice and may also help organizations, clients and the general public to see some of the differences between volunteering and professional social work (Cai et al. 2018). Linking resources is an important part of social work but remember all the different fields of practice that a social worker may work in, such as working with family violence, child protection, homelessness, disability and mental health, to name a few.*

*Developing social work skills mean that, for example, social workers may be in a better place to counsel individuals than psychologists, as both professions study counselling but social workers also understand the external impacts on the person, as well as focusing on treatment. For example, a psychologist asked me how she could help a mother who was not following a therapy program. I explained that as her son had a particular disability and health issues, the mother had to look after her son's medical needs all night and was too tired to rigorously follow cognitive behaviour therapy strategies. We worked together to link the mother to extra supports and modified the program so it would better meet her and her son's needs. Critical reflection can support social workers to be accountable (Furlong 2009) and to better understand the complex links between government, policy, organisational procedures, resourcing, theory, practice and the societal, economic and structural impacts on individuals, groups and communities.*"

Molly wrote, "Dear Hilary, but currently, what is the difference between social workers and retired mothers who have nothing to do other than mediate community conflicts? Some Chinese outsiders may feel that social workers do not need too high a qualification. As long as they have a warm heart and work training, it seems that anyone can take up the post. For example, it is still relatively normal for people with disabilities to be cared for by family members with support from community social workers who care deeply (Kwok et al. 2018).

Some students majoring in social work may be more willing to engage in higher-salary jobs, such as managers, administrators, educators and researchers, rather than social workers who work directly with the bottom of society. "

Hilary wrote, "*Dear Molly, Yes, people with warm hearts do work in supportive roles but these are not always social workers with competency-based training or tertiary education. Do these workers/volunteers understand critical thinking, can they apply theory, do they know how to communicate and take action (Zhang et al. 2019)? We also have support workers with a lived experience who undertake support work. In Australia, we call people in these roles, peer support workers (not social workers). It is very important to have peer support workers who understand what it is like to have a lived experience. Some workers then go on to study social work. There is global research outlining the role of peer support workers (Davidson et al. 2012) but this is not necessarily social work. I also wonder about our cultural differences. We need to be aware of the emerging establishment of your own social work values and ethics, for example, social harmony and traditions such as family obligations to look after elders and young children.*

*I agree that many students do not feel that social work is a profession yet. I think this is a big issue at this point in history in many countries. It is a long-term aim to change the understanding of what social work in China is (Cai et al. 2018; Jiang et al. 2019; Zhang et al. 2019). As I mentioned in an earlier email, I have ethical questions about my role in teaching an emerging profession in a culture different to my own and try to imagine*

*what social work will look like in China in the future. I try to role model inclusivity, use local examples from Wuhan and consider anti-colonial social work education. But I need your feedback and those of your Chinese lecturers to help me improve.”*

Molly wrote, “Dear Hilary, as a matter of fact, I also had some prejudice against social work when I first studied it. I still remember a conversation with my tutor a few years ago when I said that I didn't want to be a social worker after graduation. My tutor said that you can change your major in graduate school, so your major is not very important to your job search. You don't have to find a job related to your major. This conversation has puzzled me for a long time and I wonder why students who are not committed to the development of social work in China choose this major. Maybe some of them just want to get a diploma and don't care what major they choose. Maybe some of them want to study in this new field for several years and become a leading scholar. But it seems few students want to work directly as front-line social workers after graduation from university.

We know that China's social work still has a long way to go and front-line social workers are the backbone of promoting the development of social work in China. I think China needs to introduce advanced social work practice methods (Chen et al. 2018; Guo 2017; Jiang et al. 2019; Wang and Chui 2017). Furthermore, China needs to cultivate many talents who really love social work; not only to care about disadvantaged people but also to care about social workers. This could accelerate the development of social work in China.”

Hilary wrote, “*Dear Molly, I love your ideas about finding champions to care for the social workers as well. I think every country needs to do that better! Would you have a social work lecturer who might be able to talk to you about your ideas? I am also wondering why university students choose social work? A little more information about that would be helpful to better understand how the profession might develop. Do you think there is a lot of understanding about disadvantaged groups? I am wondering if high school students have much information about this before they go to university? Also why did you choose social work?*”

Molly wrote, “Dear Hilary, I'm afraid that my personal answer is not objective, so I also asked my friend some questions. He transferred his major from sociology to social work. So, at first, I thought he must want to be a social worker in the future. However, he said the reason why he transferred to social work is that he had participated in voluntary public welfare activities before and agreed with the values of social work. He likes the subject but he would not like to be a front-line social worker in the future. The reason for this is that he thinks the development of social workers in China is not yet very good and the wages are too low. But he would consider working in a non-government organisation (NGO) or a job which relates to public welfare in the future. It seems that none of my classmates would like to work as a front-line social worker after graduation but some of them want to work in a profession related to social work education, such as being a professor of social work in a university. I think some students in our major don't want to be a front-line social worker but they want to devote themselves to this field in another way.

When we were in high school, we barely had a comprehensive understanding of all the majors. Our main aim in high school is to study hard and to get high scores. Few students had time to do voluntary activities and few chances to help disadvantaged groups. I think if we knew something about a major before we went to college, it was because we had relatives or friends who were engaged in related industries around us. For example, there is no social work organisation in my hometown (my hometown is a very small city), which means that I have not had contact with any social workers before university. All I know are volunteers and neighbourhood committee aunts.

My major is largely determined by my college entrance examination scores. As for my university, the social work major has lower admission scores than other majors. I had two choices: 1) study social work at this university and minor in other majors, 2) choose a university which is not as good as this one and study a major I prefer. But this university is 211 (China's Top 50 Schools) and my parent wanted me to choose a good university rather than my preferred major.

I have thought about "why do Chinese students who major in social work do not want to be social workers after graduation?" I think many people think about this situation but so far, few people pay attention to it in China. Do Australian students who major in social work want to be social workers in the future? Perhaps we can make a comparative analysis of the employment prospects, social identity and professional education of social work in the two countries!

I don't know if this is the answer you want. As for the "would you have a social work lecturer who might be able to talk to you about your ideas?" I think most of my lecturers know about this situation, they are also working very hard on these problems and trying to teach us in a better way but sometimes it's not very easy to make changes, especially when it comes to less than ideal investment in education."

Hilary wrote, *"Dear Molly, I like your ideas—maybe you will study a PhD in the future and undertake this study?! I think we need to understand more about the challenges of being a social worker in China at this time and what makes social work a profession. As we've talked about, there is currently a lot of discussion about social work competencies, the challenges of developing tertiary education and supervisory practices, as well as adapting an originally colonial profession to your country (Cai et al. 2018; Mo et al. 2019; Morley et al. 2017; Wei and Tsui 2018; Zhang et al. 2019). Social work students in Australia do want to become social workers on graduation. There are many reasons for this such as personal experiences or a passion for social justice but I also think that the wages and the fact that social work is seen as a profession helps as well!"*

Molly wrote, "Dear Hilary, I'm very glad to think about your suggestions. If I can read a doctor's degree in the future, I'll do the comparative analysis and maybe I'll need your help ha-ha!

Nowadays, China needs a large number of professional social workers and China has invested a lot in social work education but the return is very little (Tang et al. 2017). Students majoring in social work do not want to be social workers, which is a big problem for the road to social work development in China. To be a social worker in the Chinese Mainland, there are many challenges.

I would like to share the "elite education" and "face culture" of China with you. Chinese children are taught to study hard from an early age, so that they can study in a good university, find a good job and earn a lot of money (haha, I think many countries in the world will have such a phenomenon). When relatives and friends get together on New Year's Day, they may discuss how much money they have earned in the past year or whether they have been promoted or not. It's a little strange to ask how many people you've helped this year. Although we all know to respect every profession, for example, parents tell their children to respect cleaners but parents do not want their children to be cleaners. One of my teachers said in class: "although I teach social work, I do not want my child to be a front-line social worker."

While it is a good thing that a profession can be respected by society, this does not mean that many people are willing to engage in this profession. That is, "I respect you" and "I am willing to be you" are two very different things. The former is the necessary and inadequate condition of the latter. Some people may wonder if there is a better way to contribute to social work and realise their self-worth more fully when they are employed, besides the realisation of personal value. The salary level also plays an important role in employment choices, especially for those students who study hard in order to change their social class. These are only my thoughts and I don't know if other people think the same way but I am really enjoying sharing them with you."

*3.3. The Problem of Empathy*

Molly wrote, "Dear Hilary, I have the following thoughts about empathy. You said in class that empathy is like walking in other people's footsteps. I do not think empathy is just about putting on other people's shoes to feel their feelings, because everyone's feet are different. Individuals have great differences. For example, when we look at issues with homeless people, we might have different cognitive abilities. Even when we enter other people's worlds, we also use our background filter to examine their lives, because after all, we are not them."

Hilary wrote, "*Dear Molly, yes you are right! Social workers need to imagine wearing all different shoes of all shapes and sizes. This is why social workers are trained to have the ability to use their skills to communicate across all levels of cognition and use empathy to hear what the person says in the context in which it is said. We need to use self-awareness and reflection to identify what filters we are using and why. Is it a bias, based on stigma or useful for assessment (Bassot 2016)?*"

Molly wrote, "Dear Hilary, I received your response while I was taking a psychology class. I was moved by your serious reply. You make me realise that social work is not easy work. There is no doubt that there are still a lot of problems needing to be solved in China and we are all working hard to make it better. When it comes to the relationship with clients, I thought of the following sentence: "I'm a part of your work but you are a part of my life." But it can also be that "you are a part of my work but I am a part of your life." It is not easy to explain shoe sizes (differences and boundaries!) when working with people (Howe et al. 2018).

Yesterday, our teacher asked us to choose a social work organisation for our senior internship but most students do not want to practice in the first half of the senior year, because it will take up the preparation time for the postgraduate entrance examination. But is it not for practice that we learn so much from books?

Recently, I've also read some articles on job burnout of front-line social workers in China from the perspective of emotional labour. I read that the work environment and clients, are the main factors that cause social worker's emotional burnout in the working process. The work seems to be high pressure, with lots of paperwork and I'm not sure how we put what we have learned into practice in real life (Tang et al. 2017). We might invest a lot of emotions in the clients but cannot see obvious changes. This really gives social workers a lot of emotional pressure. I think being able to remain empathic is linked to the professionalisation of social work. Burnout is very important to understand and might be why people don't want to be social workers!"

Hilary wrote, "*Dear Molly, Good timing! I've been reading articles about burnout because in Australia we often connect burnout to our emotional state. However, when we emailed earlier, we discussed how burnout and compassion fatigue (or a reduction in empathy) can be related to wage stress and a lack of professional recognition of social work (Jiang et al. 2019; Tang et al. 2017). Your comments about burnout being related to role expectations are very pertinent. This can be linked to resourcing (or a lack of resources), the value (or un-value) placed on the role, policies, expectations by clients, other workers and managers, as well as how much supervision social workers receive. Chen et al. (2018) found that supervision in field placement in China, is casual and informal, so that makes me wonder what it would be like in practice? In Australian social work education, we highly value the role of social work supervision on placement to increase critical reflective practice. Tang et al. (2017) also talk about how a lack of family recognition and support can make social workers feel discouraged. This combined with different levels of sociopolitical support and developing social work models coming from a colonial perspective can also increase the risk of burnout.*

*As I said in my first email, I'm afraid there are no easy answers. Your questions seemed at first, to be straight forward but I now realise how intertwined and unable to be separated, they are. Thank you for sharing your thoughts and helping me to understand the complexities of teaching and developing our emerging profession in your country.*"

## 4. Key Lesson Learned and Future Challenges

This a meaningful learning journey via email communication and dialogue in the teaching and learning process of a critical reflective practice course in China. Key lessons learned include teaching an emerging profession via being a role model to practice cultural-sensitivity, inclusive education and anti-colonial social work education. The teaching immersion made the lecturer aware of the cultural, political and social context of social work in China and the need to adapt and develop teaching tools and resources to meet the needs of the local students. Furthermore, being aware of the emerging establishment of social work value and ethics in China (for example, social harmony, traditional

Chinese culture); empowering individual students, groups and society to address the career barriers for social work students in China (Guo 2017). However, this discussion in critical reflective learning and teaching in China finds social work is yet to be fully developed with regards to professionalisation of the discipline, including salary and different cultural values and approaches. This dialogue did not resolve any concerns and social work is still an emerging profession in China. However, these conversations opened up an avenue to explore the questions in more depth and gave Molly and Hilary a better understanding of the concerns faced by students and the challenges in making social work an appealing and productive profession.

Further reflective questions could be: how to teach social work in developing countries or where social work is an emerging profession, particularly for educators from the western developed countries? How to teach social work to international students, who will return to their home country? How to effectively facilitate and engage social work students in critical reflection and thinking of the current structural and social issues of their home society? Where should social work educators position themselves when the pressure of social work education needs to lead to a professional job? How can we nurture the hopes of social work students and motivate them to become emerging social work practitioners? Future collaboration between academics and students in China and Australia would be helpful to progress the professionalisation of social work in China. Perhaps learnings may be transferable to and from non-Western countries where social work is still in development?

**Author Contributions:** Conceptualization, H.G., L.Y., J.L.; Methodology, H.G., L.Y., J.L.; Writing—original draft preparation, H.G., L.Y.; Writing—review and editing, H.G., L.Y., J.L.

**Funding:** This research received no external funding.

**Acknowledgments:** The authors would like to acknowledge support given by School of Human Services and Social Work, Griffith University, School of Sociology, Central China Normal University, and Australia Education Management Group in the social work teaching and research collaboration between China and Australia.

**Conflicts of Interest:** The authors declare no conflict of interest.

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
