# Peer review of "‘I Respect You but I Am Not Willing to Be You’: Critical Reflections of Western Teaching of Social Work to Students in China—What Can be Learned Both Ways?"

_socsci, doi:10.3390/socsci8100272_

Round 1
Reviewer 1 Report
Given that the article is submitted in the form of an essay, the format of presentation of the material may be acceptable.
However, there are some recommendations that partially affect the overall scientific value of the study.
The introduction does not fully reflect the problem of the study. The relevance of the study should be presented in more detail in case of issues of social work in countries which are the problem of the author's text communication. The methodology of the research is presented by very general theories and they very indirectly give an idea of the scientific basis of the presented material in the main part of the article. The findings of the study should be seriously revised. The conclusions do not reflect or address any of the questions posed by the authors during the correspondence (The wage of social workers, Professional issues, The problem of empathy) Authors should choose how to write the term burnout / burn out / burn-out (lines 351, 357, 358, 359, 361, 370)A general recommendation is to make a link between the introduction, methodology, main presentation, and conclusions of the study.
Author Response
Reviewer feedback: "The introduction does not fully reflect the problem of the study". Response to the reviewer: thank you for your suggestion, the introduction was revised to further reflect the problem of the study.Reviewer feedback: "The relevance of the study should be presented in more detail in case of issues of social work in countries which are the problem of the author's text communication. The methodology of the research is presented by very general theories and they very indirectly give an idea of the scientific basis of the presented material in the main part of the article. The findings of the study should be seriously revised. The conclusions do not reflect or address any of the questions posed by the authors during the correspondence (The wage of social workers, Professional issues, The problem of empathy)". Response to the reviewer: thank you for your suggestion. This article is based on the personal knowledge, thoughts, questions and reflections of an Australian social work lecturer and a Chinese undergraduate social work over a three-week period. We respect the value of the rigorous scientific study, but we would like to present the email conversations as an innovate way of presenting the issues of critical reflection in Western teaching of social work in China. Therefore, we may not be able to change our paper format to the traditional research paper. However, we have added a further explanation of reflection in the method, and we also consolidate the conclusion part with a more in-depth discussion of the lesson learned and future challenges of teaching social work in a non-Western context.
Reviewer feedback: "Authors should choose how to write the term burnout / burn out / burn-out (lines 351, 357, 358, 359, 361, 370)". Response to the reviewer: thank you for your suggestion, “burnout” was used consistently throughout the paper.
Reviewer feedback: "A general recommendation is to make a link between the introduction, methodology, main presentation, and conclusions of the study." Response to the reviewer: thank you for your suggestion, some transition sentences were added between different sections to strengthen the linkages.
Reviewer 2 Report
I have made some suggested edits in the manuscript itself (attached).
Here are some overall comments:
I wonder if you might consider changing the name of the article slightly. Although I think the email exchange itself demonstrates some elements of critical reflection, I don't think it is necessarily 'critical reflection' that you are teaching. The main points of the exchange are around wages, professional issues and the problem of empathy. Given this, I suggest you alter that aspect of the title to say something more like: 'critical reflections on Western teaching of social work'...
The presentation is somewhat dense (non reader-friendly). In fact, at first I thought I was going to get annoyed by the 'Dear May... Dear Kate' format. As it turned out, I didn't. However, I think it needs to be broken up a bit. Maybe you could have the emails from May in italics, and the emails from Kate in normal font. I would suggest that long slabs of text, such as that on page 2 (and elsewhere) be broken up into several paragraphs.
I think the paper has particular merit in a time when cross-cultural issues are quite prominent in public discussions (for example around such things as 'foreign influence'). Social work, as a self-proclaimed critical endeavour, needs to engage in reflection on the transferability of a highly westernised profession to other cultural contexts. We also need to consider the relevance of our western social work education to international students who will return to their home countries, perhaps, to practice.

Author Response
Reviewer's comment: "I have made some suggested edits in the manuscript itself (attached)." Response to reviewer's comment: thank you for your feedback. We have revised the highlighted points in the paper as per your suggestions. Reviewer's comment: "I wonder if you might consider changing the name of the article slightly. Although I think the email exchange itself demonstrates some elements of critical reflection, I don't think it is necessarily 'critical reflection' that you are teaching. The main points of the exchange are around wages, professional issues and the problem of empathy. Given this, I suggest you alter that aspect of the title to say something more like: 'critical reflections on Western teaching of social work'..." Response to reviewer's comment: thank you for your feedback, the title was changed according to your suggestion. Reviewer's comment: "The presentation is somewhat dense (non reader-friendly). In fact, at first I thought I was going to get annoyed by the 'Dear May... Dear Kate' format. As it turned out, I didn't. However, I think it needs to be broken up a bit. Maybe you could have the emails from May in italics, and the emails from Kate in normal font. I would suggest that long slabs of text, such as that on page 2 (and elsewhere) be broken up into several paragraphs." Response to reviewer's comment: thank you for your feedback, the email conversation was broken up (to shorter paragraphs) and italics used according to your suggestion.4.Reviewer's comment: "I think the paper has particular merit in a time when cross-cultural issues are quite prominent in public discussions (for example around such things as 'foreign influence'). Social work, as a self-proclaimed critical endeavour, needs to engage in reflection on the transferability of a highly westernised profession to other cultural contexts. We also need to consider the relevance of our western social work education to international students who will return to their home countries, perhaps, to practice." Response to reviewer's comment: we totally agree with your feedback here, and we have added a reflective question on teaching international students who will return to their home countries in the last part of the paper, which now changed from subheading "Conclusion" to "Key Lesson learned and Future Challenges". We further expanded the reflection of cross-cultural issues of teaching social work, and consolidated the main ideas and key lesson learned. Thank you again for your editing and great comments on the manuscript.